# Historical changes in wind-driven ocean circulation drive pattern of Pacific warming

Shuo Fu[1,2], Shineng Hu [✉][2], Xiao-Tong Zheng [1,3] ✉, Kay McMonigal [4,5], Sarah Larson[4] & Yiqun Tian[2]

The tropical Pacific warming pattern since the 1950s exhibits two warming centers in the western Pacific (WP) and eastern Pacific (EP), encompassing an equatorial central Pacific (CP) cooling and a hemispheric asymmetry in the subtropical EP. The underlying mechanisms of this warming pattern remain debated. Here, we conduct ocean heat decompositions of two coupled model large ensembles to unfold the role of wind-driven ocean circulation. When wind changes are suppressed, historical radiative forcing induces a subtropical northeastern Pacific warming, thus causing a hemispheric asymmetry that extends toward the tropical WP. The tropical EP warming is instead induced by the cross-equatorial winds associated with the hemispheric asymmetry, and its driving mechanism is southward warm Ekman advection due to the off-equatorial westerly wind anomalies around 5°N, not vertical thermocline adjustment. Climate models fail to capture the observed CP cooling, suggesting an urgent need to better simulate equatorial oceanic processes and thermal structures.

The tropical Pacific climate exhibits a prominent zonal asymmetry, consisting of a WP warm pool overlaid with vigorous convective rainfall and an EP cold tongue cooled by strong oceanic upwelling[1]. The pattern of tropical Pacific warming under climate change is crucial to climate sensitivity[2,3], the global hydrological cycle[4–6], and the El Niño-Southern Oscillation[7–11]. Nevertheless, how the tropical Pacific sea surface temperature (SST) has been changing in the past century and will change in the future has been under hot debate for decades (refer to the recent review by ref. 12). Climate model simulations under global warming often exhibit an equatorially enhanced warming in the eastern Pacific and therefore a weakened zonal SST gradient mimicking an El Niño-like state[13–20]. In contrast, observed SST trends in the 20th century show a narrow band of equatorial CP cooling, sometimes referred to as a La Niña-like state[21–23]. Such model-observation discrepancies persisted through the beginning of the 21st century[12,24,25]. The latest tropical Pacific SST trends since 1958 exhibit spatial structures that are more complex than a typical El Niño- or a La Niña-like

pattern (Fig. 1a; also see in ref. 25). In particular, the tropical Pacific warming pattern consists of a warm-cold-warm tripolar structure along the equator and hemispheric asymmetry in the subtropical EP.

The formation mechanisms of the recent trend pattern remain elusive given the co-existence of various physical processes. From a thermodynamic perspective, in response to increased greenhouse gases, the WP warm pool will warm less than the cold EP due to its strong background evaporative cooling that is relatively more effective in offsetting the trapped longwave radiation[13,26]. This thermodynamic effect is expected to lead to an El Niño-like warming pattern as seen in slab ocean-atmosphere coupled models[19]. When the effect of ocean circulation is considered, the climatological equatorial upwelling in the CP-WP, where the thermocline is sufficiently shallow, can act to suppress the local surface warming, while this effect is rather weak in the WP[27]. Ocean circulation changes due to surface wind changes may further contribute to the warming pattern. For example, a strengthened Walker Circulation and an intensified equatorial easterly

[1]Frontier Science Center for Deep Ocean Multispheres and Earth System (FDOMES) and Physical Oceanography Laboratory, Ocean University of China, Qingdao, China. [2]Division of Earth and Climate Sciences, Nicholas School of the Environment, Duke University, Durham, NC, USA. [3]Laoshan Laboratory, Qingdao, China. [4]Department of Marine, Earth, and Atmospheric Sciences, North Carolina State University, Raleigh, NC, USA. [5]College of Fisheries and Ocean Sciences, University of Alaska Fairbanks, Fairbanks, AK, USA. ✉e-mail: shineng.hu@duke.edu; zhengxt@ouc.edu.cn

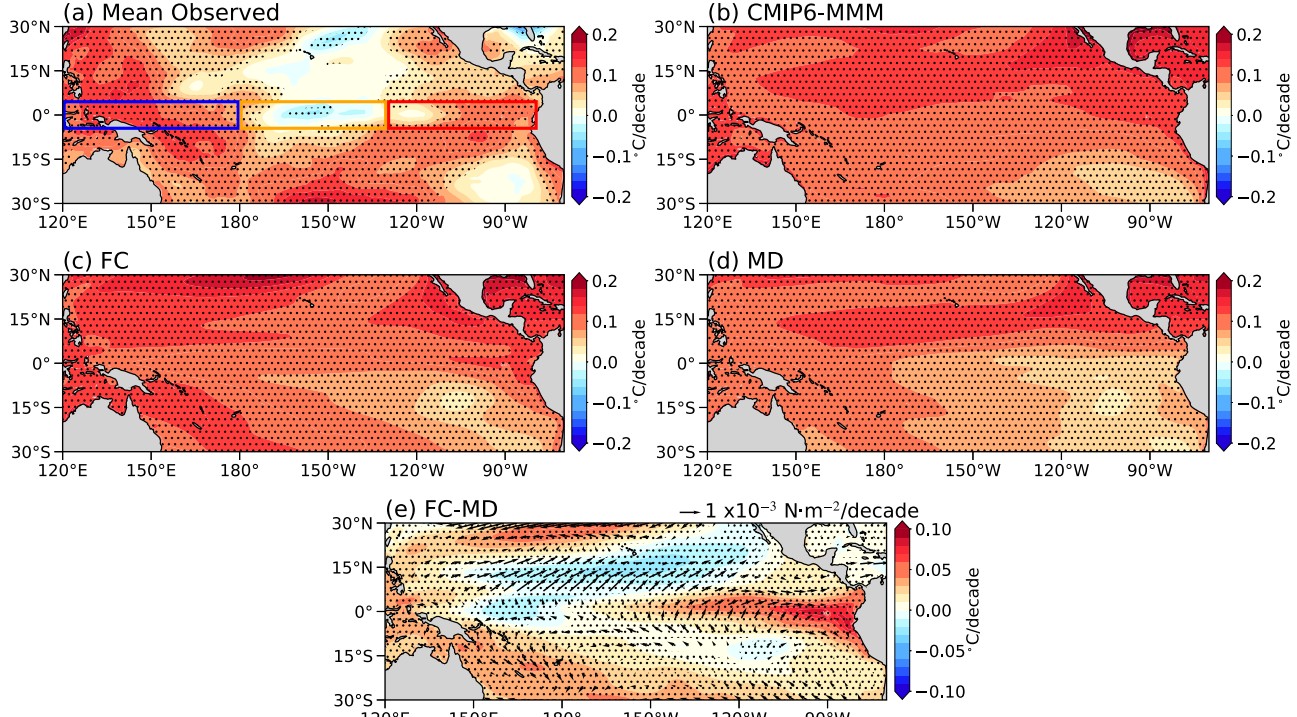

**Fig. 1 | Observed versus simulated historical tropical Pacific sea surface temperature (SST) trends. a** The mean of observed SST trend (Mean Observed) (colors; °C/decade) from five observations-based products (HadISST, COBE, COBE2, ERSSTv5, and Kaplan) over 1958–2014. **b–d** Panels showing the trend of SST (colors; °C/decade) over 1958–2014 within the CMIP6 multi-model mean (CMIP6-MMM), fully coupled ensemble mean (FC), and mechanically decoupled ensemble mean (MD), respectively. **e** Panel showing the trend of SST (colors; °C/decade) and ocean

surface wind may enhance the zonal SST gradient, further amplified by the Bjerknes feedback[28]. Some theoretical studies suggest an enhanced Walker Circulation and therefore a strengthened zonal SST gradient due to the contraction of cloud area over the western Pacific[29,30], while others imply the opposite due to energetic constraints[13,31] or low cloud-SST feedbacks[32].

What role wind-driven ocean circulation changes play in shaping the historical SST trend pattern remains poorly understood as they are often hard to fully untangle from other mechanisms. To address this question, here we compare two Community Earth System Model version 2 (CESM2) large ensembles, first introduced in ref. 33 that allow us to isolate the impacts of externally forced changes in wind-driven ocean circulations on the tropical Pacific warming pattern over the historical period. As we will show later, this unique experimental set-up and our ocean heat decomposition analysis can lead us to identify a dynamic mechanism critical to the historical tropical Pacific warming pattern that has been overlooked in the existing literature.

## Results

### Complex tropical pattern warming pattern since 1958

Firstly, we present the tropical Pacific SST trends during 1958–2014, following ref. 25, averaged across five observational datasets (see Methods; Fig. 1a; Supplementary Fig. 1). The observed warming pattern is complex, characterized by two warming centers in the WP and EP, respectively, with a distinctive cooling in the CP extending towards the east along the narrow equatorial band. Off the equator, the WP exhibits enhanced warming in both hemispheres, whereas the EP exhibits a warm-north-cold-south hemispheric asymmetry. This warming pattern results from a combination of externally forced responses (i.e. greenhouse gases, aerosols, etc.) and internal variability[1,34,35], making it hard to identify its formation mechanisms based on observations alone.

wind stress (vectors; scale bar in N m⁻²/decade) within the difference between FC and MD (FC-MD) over 1958–2014. The blue box (5°S–5°N, 120°E–180°), the orange box (5°S–5°N, 180°–130°W), and the red box (5°S–5°N, 130°W–80°W) in **a** denote the regions used to define the western Pacific (WP), central Pacific (CP), and eastern Pacific (EP), respectively. Stippling in all panels indicates significance at the 95% confidence level.

To better separate externally forced responses from internal variability, we analyze the CESM2 fully-coupled (FC), large ensemble historical simulations (see Methods). The simulations are integrated from 1850 to 2014, and the period from 1958-2014 is analyzed in our study to compare with observations. The ensemble-mean, FC CESM2-simulated SST trends can reasonably capture some of the observed warming pattern with a spatial correlation of 0.4. In particular, the observed enhanced warming in the WP and EP along the equator can be reproduced by the ensemble-mean result, suggesting an external response, although the observed CP cooling is absent in the model simulations (Fig. 1c). For individual ensemble members, the spatial correlation of the tropical Pacific SST trend pattern with observations ranges from 0.44 to −0.60. This large inter-member spread suggests that internal variability can at least explain a fraction of the model-observation discrepancies, among other factors like historical radiative forcing uncertainties and climate model biases[25]. A similar tripolar pattern is also evident in the multi-model mean historical SST trends from the Coupled Model Intercomparison Project Phase 6 (CMIP6) models, although the CP cooling is again absent (Fig. 1b). In the following analysis, we will focus particularly on the mechanisms of the enhanced WP and EP warming, and discuss the absence of CP cooling in climate models, both CESM2-LE and CMIP6, in the final discussion.

### Mechanically decoupled simulations

To investigate the impact of wind-driven ocean circulation changes, we analyze another set of large ensemble members, mechanically decoupled (MD) historical simulations with CESM2, in which ocean surface wind stress is fixed to its seasonally varying pre-industrial climatological state[33] (see Methods) using the wind stress overriding technique[36–40]. Through this approach, the Bjerknes feedback is cut off, and therefore the effects of ENSO variability and Walker Circulation

adjustment on SST trends are both eliminated[41]. We find that the MD ensemble-mean SST trends during 1958–2014 exhibit an enhanced warming spanning from the coast of Mexico to the tropical WP, resulting in a hemispheric asymmetry and an enhanced zonal SST gradient along the equator (Fig. 1d). In general, the MD ensemble mean trend agrees less well with observations than the FC ensemble mean trend with a spatial correlation of 0.25. The difference between the FC and MD represents the SST changes caused by externally-forced wind-driven ocean circulation changes, and it is characterized by an equatorial EP warming resembling an El Niño-like pattern but confined to a much narrower band within 10°S-10°N (Fig. 1e). This EP warming, combined with the WP warming in the MD (Fig. 1d), results in the two warming centers in the FC (Fig. 1c) similar to observations (Fig. 1a).

## Heat budget decomposition

To determine how wind-driven ocean circulation changes induce enhanced warming in the equatorial EP, we conduct an ocean mixed layer heat budget analysis[26,42,43] (see Methods) and decompose the SST trend ($T_s^t$) into 7 terms accounting for atmospheric adjustment and ocean dynamical processes (Fig. 2; Supplementary Fig. 2). The decomposition is summarized in Eq. (8) and the detailed methodology can be found in the Methods section.

First, the cumulative impact of these 7 factors ($T_s^t$) can accurately reproduce the simulated SST trend in the FC-MD ensemble mean (Fig. 2b; c.f. Fig. 1e) with a spatial correlation of 0.94, thereby validating our methodology. Among the 7 terms, the dominant factor for the reduction in zonal SST gradient is the ocean heat transport change ($T_{Ocn}^t$; see the quantitative decomposition in Fig. 2a), which is characterized by a strong warming confined to the eastern equatorial Pacific (Fig. 2c). It is partially offset by the shortwave radiative flux

change ($T_{SW}^t$), while other components related to atmospheric adjustment are generally weaker (Supplementary Fig. 2), including the changes in longwave radiative flux ($T_{LW}^t$), sensible heat flux ($T_{SH}^t$), and latent heat flux due to the adjustment of near-surface wind speed ($T_{LH,W}^t$), near-surface relative humidity ($T_{LH,RH}^t$) and air-sea temperature difference ($T_{LH,\Delta T}^t$). In summary, this decomposition confirms that dynamical oceanic processes dominate the wind-driven equatorial eastern Pacific warming.

Next, we further decompose the ocean mixed layer heat budget and compute 9 advective terms to elucidate specific ocean dynamical processes (see Eq. (16) in Methods). The cumulative effect of the 9 advective terms (2 linear terms and 1 nonlinear term in each direction; Supplementary Fig. 3) generally resembles the $T_{Ocn}^t$ pattern (r = 0.80), indicating that the residual term (e.g. mixing, diffusion, etc.) is relatively small (Fig. 3a; Supplementary Fig. 3). Additionally, the three nonlinear terms are generally small. Among the 6 linear terms, $-v^t \frac{\partial \bar{T}}{\partial y}$, $-\bar{u} \frac{\partial T^t}{\partial x}$, and $-\bar{v} \frac{\partial T^t}{\partial y}$ are the major contributing factors to the equatorial EP warming as are discussed further below (Fig. 3a). For $-v^t \frac{\partial \bar{T}}{\partial y}$, the off-equatorial westerly wind anomalies in the EP around 5°N induce an anomalous southward Ekman flow, which transports the warm water beneath towards the equator, thereby warming the EP cold tongue (Fig. 3b, c). This meridional warm advection is particularly strong in this region due to the strong meridional SST gradient in the EP (Fig. 3c). The climatological divergence of surface ocean currents further spread the equatorial EP warm anomaly (i.e. $-\bar{u} \frac{\partial T^t}{\partial x}$, and $-\bar{v} \frac{\partial T^t}{\partial y}$) both poleward and westward (Fig. 3d–g).

Surprisingly, vertical advection terms, and the thermocline adjustment term ($-\bar{w} \frac{\partial T^t}{\partial z}$), play only a minor role in the SST changes over the EP (Fig. 3a; Supplementary Fig. 3). The anomalous westerlies in the equatorial EP induce an anomalous downwelling that acts to warm the

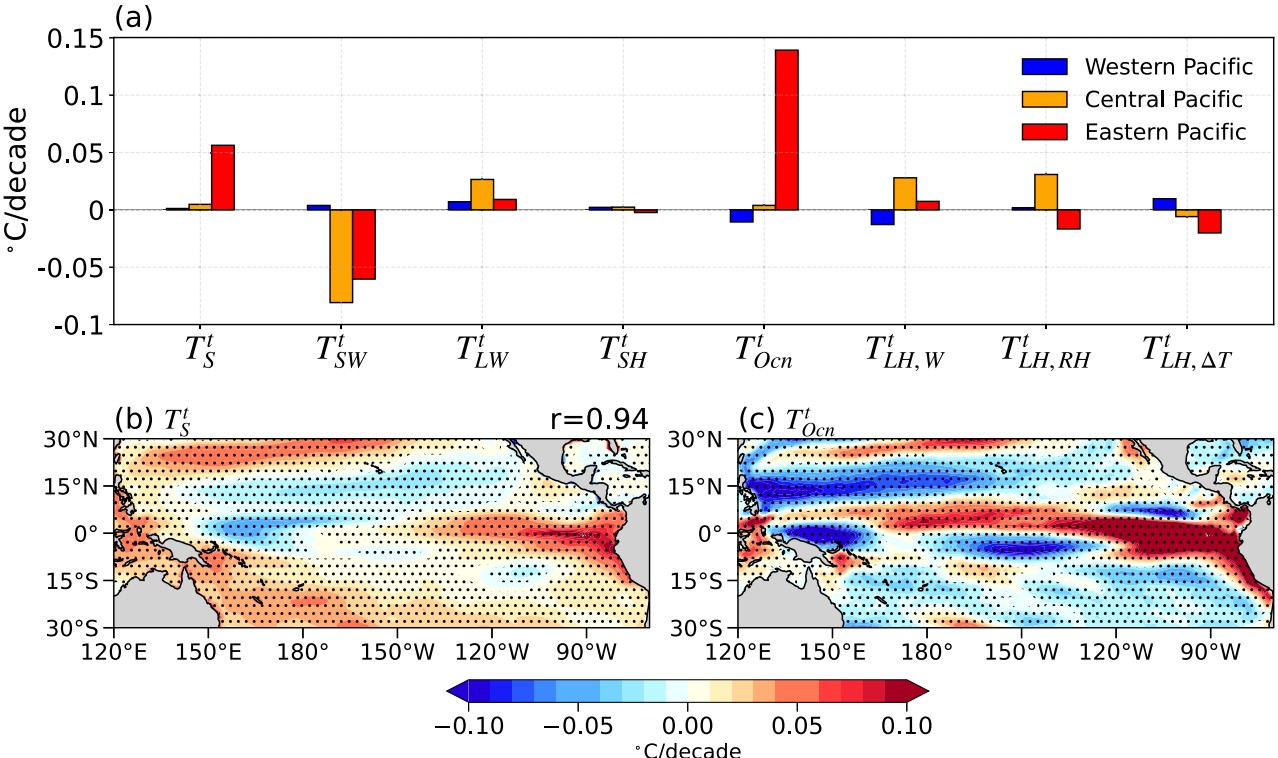

**Fig. 2 | The decomposition of sea surface temperature (SST) trend differences between fully coupled ensemble mean (FC) and mechanically decoupled ensemble mean (MD) based on ocean mixed layer heat budget. a** Bar chart showing the contribution of each component of the ocean mixed layer heat budget to FC-MD SST trends; see Methods for details. The region averaged terms consist of the western Pacific (5°S-5°N, 120°E-180°, blue box in Fig. 1a), the central Pacific (5°S-5°N, 180°–130°W, orange box in Fig. 1a), and the eastern Pacific (5°S-5°N, 130°W-80°W, red box in Fig. 1a). **b** Panel showing the SST spatial pattern of the sum of all components ($T_s^t$) and spatial correlation coefficient (r) with the FC-MD SST trend (Fig. 1e). **c** Panel showing spatial pattern of the contribution of ocean dynamics processes to SST trends ($T_{Ocn}^t$). Stippling indicates significance at the 95% confidence level.

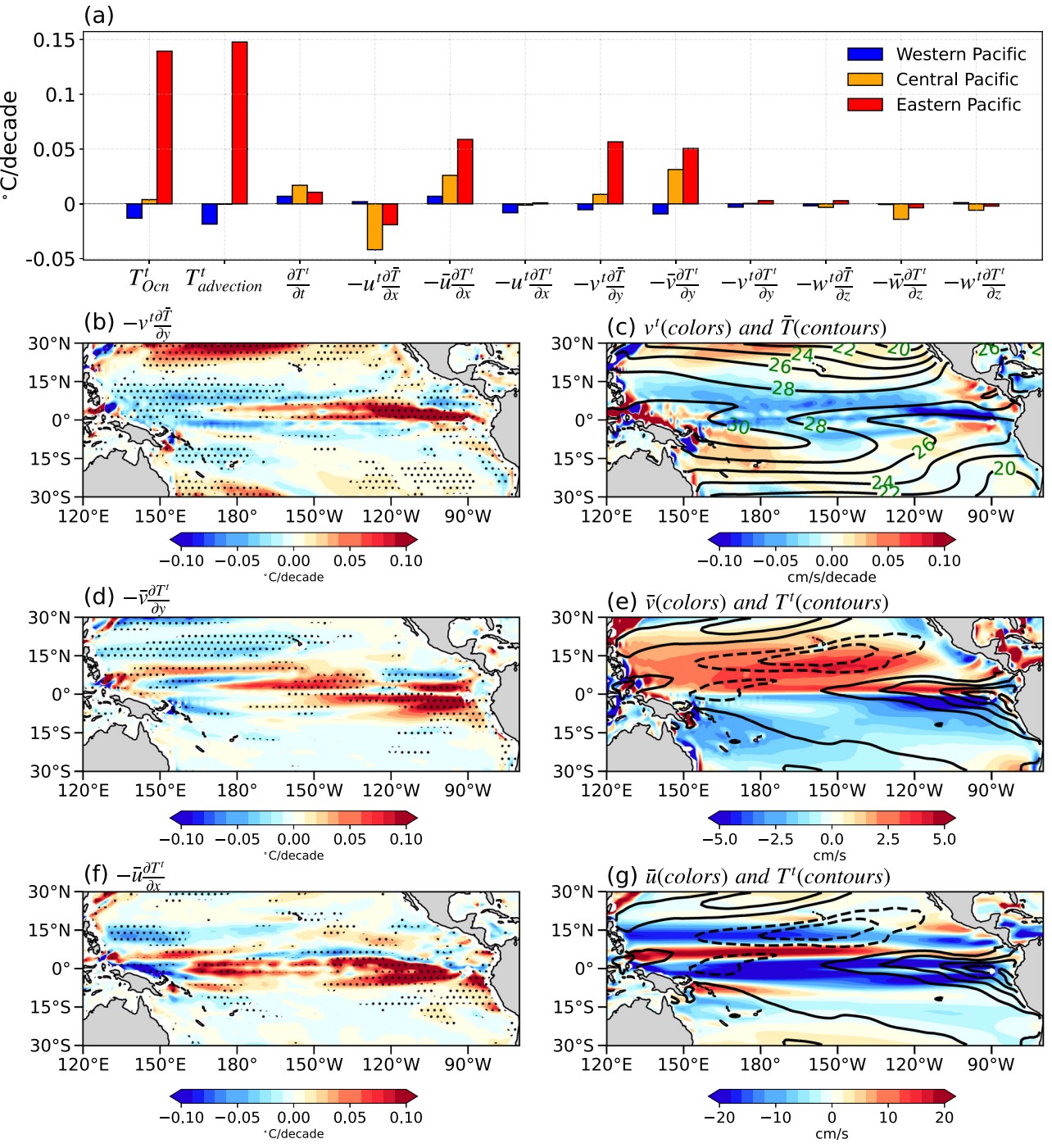

**Fig. 3 | The decomposition of ocean dynamical terms differences between fully coupled ensemble mean (FC) and mechanically decoupled ensemble mean (MD) based on the oceanic mixed-layer heat budget analysis. a** The diagnosed results of advective terms in the oceanic mixed-layer heat budget as described in Eqs. (16)–(25), with the same selected regions as in Fig. 2a; see details in Methods. $\frac{\rho_s c_p H}{\alpha LH}\frac{\partial (T^t)^f}{\partial t}$ is the mixed layer temperature tendency, here abbreviated as is $\frac{\partial T^t}{\partial t}$. **b** The spatial pattern of $-v^t\frac{\partial \bar{T}}{\partial y}$ term (unit: °C/decade). **c** Decomposing $-v^t\frac{\partial \bar{T}}{\partial y}$ into the trend of meridional advective currents term, denoted as $v^t$ (colors; cm/s/decade) and the sea surface temperature (SST) of the climatology term, represented as $\bar{T}$ (black line; unit: °C). **d** The spatial pattern of $-\bar{v}\frac{\partial T^t}{\partial y}$ term (unit: °C/decade). **e** Decomposition of

$-\bar{v}\frac{\partial T^t}{\partial y}$ into the meridional advective currents of the climatology term, denoted as $\bar{v}$ (colors; unit: cm/s) and the trend of SST term, represented as $T^t$ (the black solid line represents the positive SST trend, the black dashed line represents the negative SST trend; unit: °C/decade). **f** The spatial pattern of $-\bar{u}\frac{\partial T^t}{\partial y}$ term (unit: °C/decade). **g** Decomposition of $-\bar{u}\frac{\partial T^t}{\partial y}$ into the zonal advective currents of the climatology term, denoted as $\bar{u}$ (colors; unit: cm/s) and the trend of SST term, represented as $T^t$ (the black solid line represents the positive SST trend, the black dashed line represents the negative SST trend; unit: °C/decade). Stippling indicates significance at the 95% confidence level.

local SST ($-w^t\frac{\partial \bar{T}}{\partial z}$), but that effect is rather weak (Supplementary Fig. 3j), as the westerlies are found mostly off the equator (Fig. 1e). The climatological upwelling acts to damp the EP warming in FC-MD because of the enhanced vertical stratification ($-\bar{w}\frac{\partial T^t}{\partial z}$), but that effect is also

small as it peaks around the thermocline depth (Supplementary Figs. 3k and 4h).

By applying a similar heat budget analysis to the MD simulations (Supplementary Fig. 5), we find that the dominant mechanism for the

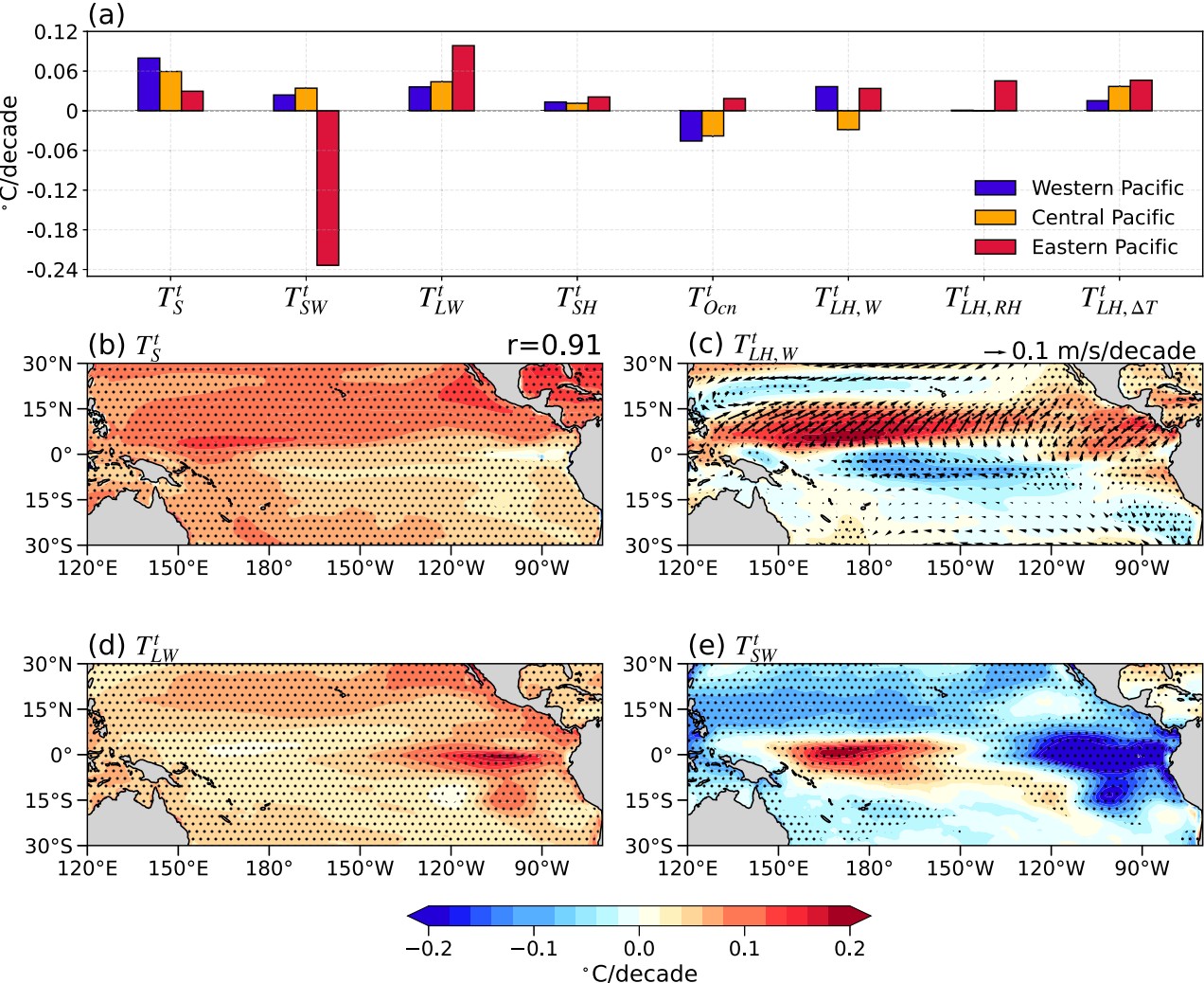

Fig. 4 | Mechanically decoupled ensemble mean (MD) sea surface temperature (SST) trend decomposition based on the ocean mixed layer heat budget analysis. a Same bar chart as Fig. 2a, but for MD. b Panel showing the SST spatial pattern of the sum of all components ($T_S^t$) and spatial correlation coefficient (r) with the MD SST trend (Fig. 1d). c Panel showing the spatial pattern of the contributions from trends in near-surface wind speed ($T_{LH,W}^t$) (colors; unit: °C/decade), with surface wind trend overlaid (arrows; unit: m/s/decade). d, e Panels showing the spatial pattern of the contribution of longwave radiative flux ($T_{LW}^t$) and surface shortwave radiative flux ($T_{SW}^t$) to SST trends (colors; unit:°C/decade), respectively. Stippling indicates significance at the 95% confidence level.

enhanced SST gradient along the equator is the shortwave radiative flux changing (Fig. 4a; Fig. 4e). An increase (decrease) in low and middle clouds leads to a cooling (warming) in the eastern (western) equatorial Pacific (Supplementary Fig. 6). In contrast, longwave radiative flux changes (i.e. greenhouse effect) induce an El Niño-like warming due partly to the "evaporative cooling" mechanism (Fig. 4d)[13], but this effect is relatively small compared to the shortwave effect. The hemispheric asymmetry of SST warming is dominated by the effect of near-surface wind speed changes (Fig. 4b, c). The cross-equatorial winds increase (decrease) the near-surface wind speed at the south (north) of the equator (Fig. 4c), thus leading to the northern hemisphere being relatively warmer.

## Discussion

Our study demonstrates that the complex warming structure in the tropical Pacific as observed since 1958 is a result of multiple physical processes. Historical radiative forcing can induce a subtropical northeastern Pacific warming extending towards the equatorial WP without invoking wind-driven ocean circulation changes. In contrast, the equatorial EP warming is driven by cross-equatorial wind anomalies in turn induced by the subtropical northeastern Pacific warming.

Interestingly, off-equatorial Ekman transport in the meridional direction turns out to be the dominant triggering mechanism for such warming. The warming effect of anomalous downwelling in the EP is relatively weak as the anomalous westerlies are located mostly off the equator, which emphasizes the recharge effect[44] due to the changes in wind stress curl instead of the Bjerknes feedback. This result highlights an overlooked role of wind-driven ocean circulation changes in the tropical Pacific warming pattern over the historical period.

Many previous studies have been devoted to understanding the cause of cross-equatorial winds in the EP, mostly related to the local or remote hemispheric asymmetry in warming as we see in our simulations[33]. The proposed mechanisms include asymmetric aerosol forcing[45,46], tropical North Atlantic warming[47], Atlantic meridional overturning circulation (AMOC)[48,49], wind–evaporation–SST feedback (WES)[26,50,51], and the Southern Ocean heat uptake[52–54]. A detailed attribution for the historical period and whether such a hemispheric asymmetry could continue into the future need to be addressed in follow-up studies.

Although the externally forced response in CESM2 can reasonably capture the equatorial WP and EP warming, it fails to reproduce the equatorial CP cooling seen in observations, like most CMIP6 models

(Fig. 1a–c). Internal climate variability could potentially explain a part of the observation-model mismatch, but the mismatch may also originate from model biases in the tropical Pacific[55–60]. For example, our heat budget analysis reveals a CP cooling effect by the climatological upwelling in FC (supplementary Fig. 4b), consistent with the thermostat mechanism[27], but this effect is rather weak in CESM2. A model bias in the magnitude of equatorial Pacific upwelling, in turn driven by surface winds, can potentially lead to a bias in the upwelling-induced CP cooling rate, and needs to be carefully assessed in future investigations. Other model biases, including the double ITCZ and the cold tongue biases, and their potential impacts on the tropical Pacific warming pattern also need to be addressed.

## Methods

### Observational data sets
We use five monthly observational SST datasets: (1) National Oceanic and Atmospheric Administration Extended Reconstructed Sea Surface Temperature Version 5 (ERSSTv5) with resolution $2° × 2°$[61]. (2) Hadley Centre Sea Ice and SST v.1.1 (HadISST 1.1) with resolution $1° × 1°$[62]. (3) Centennial In Situ Observation-Based Estimates of the Variability of SST and Marine Meteorological Variables (COBE) with resolution $1° × 1°$[63], and (4) COBE2 with resolution $1° × 1°$[64]. (5) Kaplan Extended SST v2 with resolution $5° × 5°$[65]. In this study, the period from 1958 to 2014 has been chosen for calculating the SST trends. We chose 1958 as the starting year by following refs. [1,25], which was determined according to the availability of ECMWF/ORAS4 ocean reanalysis data, and we chose 2014 as the ending year because our mechanically decoupled experiments followed the standard of the CMIP6 historical simulations that ended in the year 2014.

### CMIP6 model data
We examined the SST trend in 31 climate models from the Coupled Model Intercomparison Project Phase 6 (CMIP6). We chose the 1958–2014 interval in historical simulations to compute the trend and compared it with the trend of the observed data set. The 31 climate models are ACCESS-CM2, ACCESS-ESM1-5, CAMS-CSM1-0, CanESM5, CanESM5-CanOE, CESM2, CESM2-FV2, CESM2-WACCM, CESM2-WACCM-FV2, CNRM-CM6-1, CNRM-CM6-1-HR, CNRM-ESM2-1, E3SM-1-0, E3SM-1-1, E3SM-1-1-ECA, EC-Earth3, EC-Earth3-Veg, FGOALS-f3-L, HadGEM3-GC31-LL, HadGEM3-GC31-MM, IPSL-CM6A-LR, MIROC-ES2L, MIROC6, MPI-ESM1-2-HR, MPI-ESM1-2-LR, MRI-ESM2-0, NESM3, NorESM2-LM, NorESM2-MM, SAM0-UNICON, UKESM1-0-LL.

### Large ensemble historical simulations
In this study, we analyze two large ensemble coupled climate model simulations using the Community Earth System Model version 2 (CESM2)[66,67]. Two CESM2 ensembles both have nominally 1° horizontal resolution and are forced with realistic, time-varying 1850-2014 external forcing, including greenhouse gasses, natural and anthropogenic aerosol emissions, and solar radiation. Smoothed biomass burning forcing was used for these simulations[68].

The first CESM2 ensemble, referred to as FC for "Fully Coupled Model", is created by branching 50 ensemble members from a pre-industrial model run for over 1000 years by NCAR. Ten of the ensemble members are branched from different pre-industrial control simulation dates (macro ensemble). The other 40 ensemble members are created by branching from four pre-industrial control simulation starting dates and adding a random perturbation to the atmospheric potential temperature field to create 10 micro ensemble members per branch date[68]. These runs are publicly available as part of the CESM2 Large ensemble. In the FC ensemble, the ocean and atmosphere exchange buoyancy fluxes and wind stress.

Wind stress overriding experiments have been demonstrated to be helpful in isolating the influence of wind-driven ocean circulation on the ocean-atmosphere coupled system[36–40]. The second CESM2

ensemble, MD for "Mechanically Decoupled Model", is created by branching 20 members from a pre-industrial MD model run[33]. Each ensemble member is branched from a different pre-industrial control date, making it a macro ensemble. The 6-hourly climatological wind stress forcing in the MD ensemble is calculated from 50 years of hourly wind stress output from an FC pre-industrial run. The ocean and atmosphere exchange time-varying buoyancy fluxes, but the ocean is forced by a seasonally-varying fixed wind stress climatology, calculated from pre-industrial conditions. More details can be found in ref. [33].

### Ocean mixed layer heat budget analysis
To further understand the mechanisms of wind-driven ocean circulation changes that induce an eastern equatorial Pacific warming, we analyzed the ocean mixed layer heat budget following previous studies[42,43]. The ocean surface mixed-layer budget equation is:

$$\rho_s c_p H \frac{\partial SST}{\partial t} = F_{SW} + F_{LW} + SH + LH + Ocn \tag{1}$$

Here, the left-hand side represents the mixed-layer heat storage term, where $\rho_s$ is the ocean density, $c_p$ is the specific ocean heat, $H$ is the time-varying mixed layer depth from the model, and $SST$ is the mixed-layer temperature. The right-hand side consists of net surface shortwave ($F_{SW}$), longwave ($F_{LW}$) fluxes, sensible heat fluxes ($SH$) and latent heat fluxes ($LH$), and heat flux due to ocean dynamics ($Ocn$). Here we define downward as positive.

When we take the linear trend of Eq. (1), the left-hand side which represents the trend of heat storage is negligible[26,69], which is written as follows:

$$0 = SW^t + LW^t + SH^t + LH^t + Ocn^t \tag{2}$$

the superscript t denotes linear trends from 1958 to 2014. The latent heat flux term is directly related to $SST$ via saturation vapor pressure:

$$LH = L_v c_e \rho_{air} W [q_s(SST) - q_a] \tag{3}$$

where $q_a$ is the specific humidity of the air above the sea surface, $L_v$ is the latent heat of vaporization, $\rho_{air}$ is the density of the air and $W$ is near-surface wind speed. The $q_a$ term can be expressed as:

$$q_a = RH_0 q(SST - \Delta T) \tag{4}$$

where $RH_O$ is the relative humidity at the sea surface, $\Delta T$ is the temperature gradient near the sea surface, which is defined as $(SST - Ta)$. Using the Clausius-Clapeyron equation, Eq. (4) can be expressed as:

$$q_a = RH_0 q_s(SST) e^{\alpha \Delta T} \tag{5}$$

where $\alpha = \frac{L_v}{R_v T^2} \approx 0.06\,K^{-1}$. We can substitute Eq. (5) into Eq. (3) and obtain a new expression for the latent heat flux:

$$LH = -L_v c_e \rho_{air} W (1 - RH0 e^{\alpha \Delta T}) q_s(T_s) \tag{6}$$

Then, we use linear regression to get the linear trend of latent heat flux from Eq. (6):

$$LH^t = \frac{\partial LH}{\partial SST} SST^t + \frac{\partial LH}{\partial W} W^t + \frac{\partial LH}{RH_0} RH_0{}^t + \frac{\partial LH}{\Delta T} \Delta T^t$$
$$= \alpha \overline{LH} SST^t + \frac{\overline{LH}}{\overline{W}} W^t - \frac{\overline{LH}}{e^{\alpha \overline{\Delta T}} - \overline{RH_0}} RH^t + \frac{\alpha \overline{LH}\,\overline{RH}}{e^{\alpha \overline{\Delta T}} - \overline{RH_0}} \Delta T^t \tag{7}$$

where the $\overline{LH}, \overline{W}, \overline{\Delta T}, \overline{RH_0}$ is climatology computed using the period of 1958-2014. Then we rewrite Eq. (2) as a diagnostic equation of the SST

trend to show how different forcing terms contribute to the SST trend:

$$T_s^t = -\frac{F_{SW}^t + F_{LW}^t + SH^t + Ocn^t + LH_W^t + LH_{RH}^t + LH_{\Delta T}^t}{\alpha \overline{LH}}$$
$$= T_{SW}^t + T_{LW}^t + T_{SH}^t + T_{LH,W}^t + T_{LH,RH}^t + T_{LH,\Delta T}^t + T_{Ocn}^t \quad (8)$$

We represent $SST^t$ as $T_s^t$ in the equation, the latent heat term can be further broken down into contributions from trends in near-surface wind speed ($T_{LH,W}^t$), near-surface relative humidity ($T_{LH,RH}^t$), and air-sea temperature gradient ($T_{LH,\Delta T}^t$), where the right-hand side terms can be calculated as follows Eqs. (9)–(15).

$$T_{SW}^t = -\frac{F_{SW}^t}{\alpha \overline{LH}} \quad (9)$$

$$T_{LW}^t = -\frac{F_{LW}^t}{\alpha \overline{LH}} \quad (10)$$

$$T_{SH}^t = -\frac{F_{SH}^t}{\alpha \overline{LH}} \quad (11)$$

$$T_{Ocn}^t = -\frac{F_{Ocn}^t}{\alpha \overline{LH}} \quad (12)$$

$$T_{LH,w}^t = -\frac{LH_w^t}{\alpha \overline{LH}} = -\frac{W^t}{\alpha \overline{W}} \quad (13)$$

$$T_{LH,RH}^t = -\frac{LH_{RH}^t}{\alpha \overline{LH}} = \frac{RH_0^t}{\alpha(e^{\alpha \overline{\Delta T}} - \overline{RH_0})} \quad (14)$$

$$T_{LH,\Delta T}^t = -\frac{LH_{\Delta T}^t}{\alpha \overline{LH}} = -\frac{\overline{RH_0}}{e^{\alpha \overline{\Delta T}} - \overline{RH_0}} \Delta T^t \quad (15)$$

Here, the cumulative of these 7 factors ($T_s^t$) may not exactly match with the model-simulated SST due to the assumptions made during the derivation process.

## Oceanic mixed-layer heat budget Equations

To understand the roles of ocean dynamical processes in causing the eastern equatorial Pacific warming, we further decompose the influence of ocean heat transport changes as follows:

$$T_{advection}^t = \frac{\rho_s c_p H}{\alpha \overline{LH}}\left[\left(-u'\frac{\partial \overline{T}}{\partial x}\right)^t + \left(-\bar{u}\frac{\partial T'}{\partial x}\right)^t + \left(-u'\frac{\partial T'}{\partial x}\right)^t + \left(-v'\frac{\partial \overline{T}}{\partial y}\right)^t \right.$$
$$+ \left(-\bar{v}\frac{\partial T'}{\partial y}\right)^t + \left(-v'\frac{\partial T'}{\partial y}\right)^t + \left(-w_b'\frac{\partial(\overline{T-T_b})}{H}\right)^t$$
$$\left. + \left(-\bar{w}_b\frac{\partial(T-T_b)'}{H}\right)^t + \left(-w_b'\frac{\partial(T-T_b)'}{H}\right)^t\right] + R$$
$$(16)$$

where the overbar denotes the climatological mean state, the prime denotes anomaly, and the superscript '$t$' denotes long-term trend. Each zonal, meridional, and vertical ocean 3-dimensional advection term consists of three components, two linear terms, and one nonlinear term. $T_{advection}^t$ is the sum of the terms on the right-hand side of Eq. (16), excluding the residual terms ($R$), which means that the residual terms are not as important for the heat budget at the mixed layer depth. We use $T_{advection}^t$ for comparison with $T_{Ocn}^t$. $T$ is the mixed-layer average temperature, $T_b$ is the bottom of mixed-layer temperature, $u$, $v$, and $w$ represent 3D mixed-layer current velocity, $w_b$ is the bottom of mixed-layer temperature and current velocity, and $H$ denotes the climatological seasonally varying mixed-layer depth in the

equatorial Pacific, which is located at a 0.5 °C difference from the surface SST. $R$ is the residual term. The term $\frac{\rho_s c_p H}{\alpha \overline{LH}}$ varies in space much less than those advective terms. The climatological annual cycle is calculated based on the period from 1958–2014. The anomaly field is calculated by subtracting the monthly mean field from its climatological annual cycle at each period respectively.

For clarity, the terms on the right-hand side of Eq. (16) are abbreviated in the figures as follows,

$$-u^t\frac{\partial \overline{T}}{\partial x} = \frac{\rho_s c_p H}{\alpha \overline{LH}}\left(-u'\frac{\partial \overline{T}}{\partial x}\right)^t \quad (17)$$

$$-\bar{u}\frac{\partial T^t}{\partial x} = \frac{\rho_s c_p H}{\alpha \overline{LH}}\left(-\bar{u}\frac{\partial T'}{\partial x}\right)^t \quad (18)$$

$$-u^t\frac{\partial T^t}{\partial x} = \frac{\rho_s c_p H}{\alpha \overline{LH}}\left(-u'\frac{\partial T'}{\partial x}\right)^t \quad (19)$$

$$-v^t\frac{\partial \overline{T}}{\partial y} = \frac{\rho_s c_p H}{\alpha \overline{LH}}\left(-v'\frac{\partial \overline{T}}{\partial y}\right)^t \quad (20)$$

$$-\bar{v}\frac{\partial T^t}{\partial y} = \frac{\rho_s c_p H}{\alpha \overline{LH}}\left(-\bar{v}\frac{\partial T'}{\partial y}\right)^t \quad (21)$$

$$-v^t\frac{\partial T^t}{\partial y} = \frac{\rho_s c_p H}{\alpha \overline{LH}}\left(-v'\frac{\partial T'}{\partial y}\right)^t \quad (22)$$

$$-w^t\frac{\partial \overline{T}}{\partial z} = \frac{\rho_s c_p H}{\alpha \overline{LH}}\left(-w_b'\frac{\partial(\overline{T-T_b})}{H}\right)^t \quad (23)$$

$$-\bar{w}\frac{\partial T^t}{\partial z} = \frac{\rho_s c_p H}{\alpha \overline{LH}}\left(-\bar{w}_b\frac{\partial(T-T_b)'}{H}\right)^t \quad (24)$$

$$-w^t\frac{\partial T^t}{\partial z} = \frac{\rho_s c_p H}{\alpha \overline{LH}}\left(-w_b'\frac{\partial(T-T_b)'}{H}\right)^t \quad (25)$$

## Trends and significance

All the linear trends are calculated by applying a linear least-squares regression model to the spatially integrated time series. To determine the regions and time periods where trends are significantly different, we consider the spread of the ensemble members as a normal distribution. To test if the distributions are significantly different, we calculate the Z statistic as follows Eq. (26) and use 95% significance ($Z \geq 1.96$) as a threshold, where:

$$Z = \frac{X_{FC} - X_{MD}}{\sqrt[2]{\sigma_{FC}^2 + \sigma_{MD}^2}} \quad (26)$$

X is the ensemble mean from each experiment. with shading showing the 95% confidence interval. $\sigma$ is the standard deviation of each ensemble member divided by the square root of the number of ensemble members.

## Data availability

For observational datasets, the NOAA's ERSSTv5 data are available at https://psl.noaa.gov/data/gridded/data.noaa.ersst.v5.html; HadISST 1.1 data at https://www.metoffice.gov.uk/hadobs/hadisst/data/download.html; COBE SST at https://psl.noaa.gov/data/gridded/data.cobe.html; COBE2 SST at https://psl.noaa.gov/data/gridded/data.

cobe2.html; Kaplan Extended SST v2 at https://psl.noaa.gov/data/gridded/data.kaplan_sst.html; CMIP6 data at https://esgf-node.llnl.gov/search/cmip6. CESM2 Fully-Coupled (FC) Large ensemble members at https://www.cesm.ucar.edu/projects/community-projects/LENS2/. Select CESM2 Mechanically Decoupled (MD) pre-industrial and used Large ensemble data is found at https://www.earthsystemgrid.org/dataset/ucar.cgd.cesm2.mdpc.html and https://zenodo.org/records/10484207.

## Code availability
Codes for the main results are available on Zenodo at https://zenodo.org/records/10494999.

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

## Acknowledgements

S.F. and X.-T.Z. are supported by the National Natural Science Foundation of China (41975092, 42230405) and the Taishan Scholars Project of Shandong Province (No. tsqn202306095). S.F. is supported by the China Scholarship Council (202106330039). S.H. acknowledges the computational support by the Dean's Equipment Betterment Fund, Nicholas School of the Environment, Duke University. K.M. and S.L. are supported by NSF Grant AGS-1951713. We acknowledge the high-performance computing support from Cheyenne provided by NCAR's Computing and Information Services Lab, sponsored by NSF. Computational resources at the NCAR-Wyoming Supercomputing Center and the Duke Computing Center are also acknowledged.

## Author contributions

S.F., S.H., and X.-T.Z. conceived the original idea and wrote the first draft of the manuscript. S.L. and K.M. conducted the CESM2 mechanically decoupled simulations. S.F. performed the analysis and generated the figures. Y.T. contributed to the ocean heat budget decomposition analysis. S.F., S.H., X.-T.Z, S.L., K.M., and Y.T. contributed to the interpretation of results and edited the manuscript.

## Competing interests

The authors declare no competing interests.
