## [Peer Review File · Nature Communications]

Historical changes in wind-driven ocean circulation drive pattern of Pacific warmingREVIEWER COMMENTS

Reviewer #1 (Remarks to the Author):

Fu and coauthors explore the drivers of patterns of warming in the tropical Pacific using a novel modelling approach that isolates oceanic responses to wind changes. They find that a hemispheric asymmetry is the main driver initiating a positive feedback loop producing enhanced equatorial warming – an ubiquitous pattern in climate projects performed by models used by the IPCC. The use of a novel modeling approach, namely the Mechanical Decoupling, warrants publication, however a few details must be clarified before I can recommend this study for publication.

Main comments:

Title: I suggest a more declarative title, such as: “Cross-equatorial wind response drives pattern of warming in climate models”. This title is just a suggestion, I think a paper in a high impact journal will benefit from a more specific title.

A rationale for the interval chosen to analyze the forced trends needs to be provided. Is this the interval when the model best agrees with observed changes? Or is this the interval when we expect the stronger externally forced responses?

Line 170: What is surprising about a muted vertical advection response? The authors should elaborate on this statement. I am intrigued that the FC-MD difference shows very little influence from the change in vertical advection by the change in vertical currents ($-\Delta(\bar{w}) \cdot dT/dz$) – where Δ is the change in response to external forcing. The change in vertical advection by the change in vertical stratification ($-\bar{w} \cdot \Delta(dT/dz)$) should produce cooling because the tendency towards westerly winds should shoal thermocline across the equatorial Pacific (the discharge effect). I suggest analyzing these changes in both MD and in the FC-MD response. Both should show changes in dT/dz and FC-WD should show a decrease in upwelling. I think these changes need to be analyzed. Perhaps looking at the changes in the (lon-depth) plane could help see how a reduction in upwelling and a shoaling of the thermocline affect occur as a function of longitude and depth.

Line 195-196: “...to be the dominant triggering mechanism for such warming, rather than a thermocline deepening as one would expect from the Bjerknes feedback.” The authors seem to ignore that the equilibrated thermocline response to weaker easterly winds does NOT involve a deepening in the central and eastern Pacific, instead it involves a zonal mean shoaling – the discharge effect. Thus the Bjerknes feedback on these timescales does not involve the thermocline and is dominated by wind-driven changes in currents – both horizontal and vertical.

Discussion: What is the impact of systematic model errors in the mechanism presented in this study? I think the authors need to discuss (and analyze) the seasonality of the response to determine whether it arises from the double ITCZ error or not.

Fig. 3. Something is odd about some labels in Fig. 3. I don't understand why 4 panels are shown for each advective term. I can imagine showing 3 panels, what is the 4th one? Also, the labels do not seem straightforward to understand. Please revise so it is easier to digest what each map is showing.

Reviewer #2 (Remarks to the Author):

This study investigates the historical tropical Pacific warming pattern since 1958, characterized by warming in the eastern and western Pacific and cooling in the central tropical Pacific, as well as a hemispheric asymmetry in the subtropical eastern Pacific. Authors use the mechanical decoupled simulations to isolate the impact of wind-driven ocean circulations, which allow them to investigate the impact of dynamic and thermodynamic factors on the historical tropical warming pattern. They find that eastern tropical Pacific warming is related to southward Ekman transport induced by anomalous off-equatorial westerly winds. While the warming in the western tropical Pacific is attributed to heat transported by asymmetric equatorial surface winds from the northeastern Pacific, induced by historical external forcing. This paper is well written and provides important implications for the mechanisms of the observed tropical Pacific warming, particularly for the eastern tropical Pacific warming. Their proposed mechanism differs from previous studies, which suggested that the warming in the eastern tropical Pacific is caused by the deepening of the thermocline as a result of Bjerknes feedback. Overall, I think this paper is suitable for the scope of Nature Communications and I would recommend the acceptance of this manuscript once the suggested comments have been adequately addressed.

I have provided detailed comments below in my major and specific comments. My main concern is that authors highlight the observed tripolar Pacific warming pattern, including the weak cooling in the central tropical Pacific, however, authors don't give sufficient elaboration on the results about central Pacific in observations and models. Main text focus on western and eastern tropical Pacific warming. All bar charts include data for the central tropical Pacific, but the main text does not provide a comprehensive description or discussion of these results. Also, I would suggest authors to provide more information on the robustness of SST trends across observation datasets, especially SST trends over central tropical Pacific.

Sincerely,

A Reviewer for Nature Communications

Major Comments:

1. The manuscript prominently highlights the observed tripolar warming pattern. However, models fail to reproduce the observed weak cooling in the central tropical Pacific, and authors do not provide a comprehensive explanation for results of central tropical Pacific shown in the figures. Main text focuses

on explaining the causes of symmetric warming in the western tropical Pacific and asymmetric warming in the eastern tropical Pacific. There hasn't been sufficient explanation regarding the possible cause of the observed central tropical Pacific cooling (the tripolar warming pattern) and the discrepancy between models and observations. If the main focus is on the western and eastern tropical Pacific warming pattern, authors might consider adjusting the related context. Another point is that the time span of model simulations and observations are not the same. It is better to be consistent with the study period.

Specific Comments:

1. In Fig.1a, the cooling over the central tropical Pacific and southeastern Pacific appears weak and statistically insignificant. Do all observation datasets show this cooling? Also, considering that the spatial resolution of observation datasets varies, it would be helpful to clarify whether all datasets are interpolated to the same resolution. Because interpolation could impact the estimate of observed SST trends and, therefore, their average, especially when dealing with datasets like Kaplan Extended SST v2, which has a coarser grid compared to others. It would be helpful to show SST trend maps for each observation dataset. An alternative way is to show the agreement in the sign of SST trends among observation datasets. Maybe include more observation datasets can increase confidence on the sign of SST change over central tropical Pacific.
2. Is the result sensitive to how the eastern and western tropical Pacific regions is defined? In Fig. 1c, the weaker warming over eastern tropical Pacific extends to central tropical Pacific in CEMS2 LENS. Also, the warming over eastern tropical Pacific extends to central tropical Pacific driven by externally forced winds in Fig. 1e.
3. Line 68 has a superscript stop.

Response to the reviews of the manuscript

“Unfolding the role of wind-driven ocean circulation in the historical Pacific warming pattern”

By Shuo Fu, Shineng Hu, Xiao-Tong Zheng, Kay McMonigal, Sarah Larson, Yiqun Tian

We thank the reviewers for their comments and suggestions that led to many improvements in the paper. The text was further edited for misprints and clarity. We feel that the revised draft is greatly improved thanks to the reviewers' constructive comments. Our point-by-point responses to the reviews are below (reviewers' comments in italics, authors' responses in blue font).

Reviewer #1 (Remarks to the Author):

Fu and coauthors explore the drivers of patterns of warming in the tropical Pacific using a novel modelling approach that isolates oceanic responses to wind changes. They find that a hemispheric asymmetry is the main driver initiating a positive feedback loop producing enhanced equatorial warming – an ubiquitous pattern in climate projects performed by models used by the IPCC. The use of a novel modeling approach, namely the Mechanical Decoupling, warrants publication, however a few details must be clarified before I can recommend this study for publication.

Main comments:

Title: I suggest a more declarative title, such as: “Cross-equatorial wind response drives pattern of warming in climate models”. This title is just a suggestion, I think a paper in a high impact journal will benefit from a more specific title.

Yes, this is a very helpful suggestion. We agree with the reviewers that a more specific title would be useful. We have changed the title to "Historical changes

in wind-driven ocean circulation drive pattern of Pacific warming" (Lines 1-2).

A rationale for the interval chosen to analyze the forced trends needs to be provided. Is this the interval when the model best agrees with observed changes? Or is this the interval when we expect the stronger externally forced responses?

We chose 1958 as the starting year by following Seager et al. (2019, 2022), which was determined according to the availability of ECMWF/ORAS4 ocean reanalysis data. We chose 2014 as the ending year because our mechanical decoupling experiments followed the standard of the CMIP6 historical simulations that ended in the year 2014 (Lines 239-244). It is also worth noting that the warm-cold-warm tripolar structure observed during 1958-2014 is also seen for other periods, for example, 1900-2014, 1930-2014, and 1958-2022 (Fig. R1).

Fig. R1 | a, b, c represent the SST trend from observations-based products during 1900–2014, 1930–2014, and 1958–2022, respectively. Stippling in all panels indicates significance at the 95% confidence level.

*Line 170: What is surprising about a muted vertical advection response? The authors should elaborate on this statement. I am intrigued that the FC-MD difference shows very little influence from the change in vertical advection by the change in vertical currents $(-\Delta(\bar{w}) * dT_{bar}/dz)$ – where Δ is the change in response to external forcing. The change in vertical advection by the change in vertical stratification $(-\bar{w} * \Delta(dT_{bar}/dz))$ should produce cooling because the tendency towards westerly winds should shoal thermocline across the equatorial Pacific (the discharge effect). I suggest analyzing these changes in both MD and in the FC-MD response. Both should show changes in dT/dz and FC-WD should show a decrease in upwelling. I think these changes need to be analyzed. Perhaps looking at the changes in the (lon-depth) plane could help see how a reduction in upwelling and a shoaling of the thermocline affect occur as a function of longitude and depth.*

Thanks for raising this important point. In the previous version of the manuscript, we didn't discuss the two vertical advective terms in detail because their contributions were relatively small (Fig. 3a). Now we agree with the reviewer that understanding why those terms are small will be very helpful. Following the reviewer's suggestion, we have now plotted both the mean state and the change in temperature and vertical velocity as a function of depth and longitude in the equatorial Pacific (Fig. R2).

- (1) Indeed, the eastern equatorial Pacific westerly anomalies in FC lead to an anomalous downwelling (i.e. weaker upwelling), which is absent in MD wherein the wind stress anomalies are suppressed (left column in Fig. R2). But this effect peaks around 100 m and is rather weak at the base of mixed layer depth. Therefore, the vertical advective term due to the change in

upwelling ($-w^t \frac{\partial \bar{T}}{\partial z}$) is weakly positive in the eastern Pacific in FC-MD.

- (2) For temperature changes, we find that the change in wind stress curl tends to recharge the equatorial western Pacific but has little effect on the eastern equatorial Pacific as the cross-equatorial westerlies are located mostly off the equator (Fig. R3; cf. Fig. 1e). As a result, thermocline depth, here defined as the 20°C isotherm, gets slightly deeper in the western Pacific but stays almost unchanged in the eastern Pacific. Despite the small change in thermocline depth in general, the vertical stratification increases in the equatorial Pacific in FC-MD (i.e., $\frac{\partial T^t}{\partial z}$). Therefore, the vertical advective term due to the change in temperature structure ($-\bar{w} \frac{\partial T^t}{\partial z}$) is negative, but its amplitude is also small (Fig. 3a) because the stratification change is found mostly at the thermocline depth (middle column in Fig. R2).
- (3) The nonlinear vertical advective term is shown in the right column in Fig. R2, and its resultant amplitude is also small (Fig. 3a).

In summary, we find that ocean subsurface temperature and upwelling changes are consistent with our heat budget analysis. We find this in-depth analysis, as suggested by the reviewer, quite informative, and have now added relevant discussions at Lines 173-181 and the new figures to Supplementary Fig. 4.

Also, we note that the model representation in the vertical structure of temperature and upwelling could potentially contain biases, and the model-observation mismatch in the equatorial central Pacific cooling is possibly related to this. We have now added a discussion to this point at Lines 217-227, although it is out of the scope of this study.

Fig. R2 | Left columns represent the decomposition of the $-w^t \frac{\partial \bar{T}}{\partial z}$ terms in FC, MD, and FC-MD, respectively, the solid line represents the mean state of temperature structure, and the color represents the trend of vertical velocity. Middle columns represent the decomposition of the $-\bar{w} \frac{\partial T^t}{\partial z}$ terms in FC, MD, and FC-MD, respectively, the solid (dashed) line represents the positive (negative) trend of temperature structure, and the color represents the mean state of vertical velocity. Right columns represent the decomposition of the $-w^t \frac{\partial T^t}{\partial z}$ term in FC, MD, and FC-MD, respectively, the solid (dashed) line represents the positive (negative) trend of temperature structure, and the color represents the trend of vertical velocity. The purple line represents the mixed layer depth, and the green solid (dashed) line represents the thermocline depth in FC (MD).

Fig. R3 | The changes in the trend of wind stress curl during 1958-2014 in FC-MD. Stippling in the panel indicates significance at the 95% confidence level.

Line 195-196: “...to be the dominant triggering mechanism for such warming, rather than a thermocline deepening as one would expect from the Bjerknes feedback.” The authors seem to ignore that the equilibrated thermocline response to weaker easterly winds does NOT involve a deepening in the central and eastern Pacific, instead it involves a zonal mean shoaling – the discharge effect. Thus the Bjerknes feedback on these timescales does not involve the thermocline and is dominated by wind-driven changes in currents – both horizontal and vertical.

Thanks for pointing this out. Our previous statement is indeed not accurate. We have now rephrased this sentence (Lines 203-206) by emphasizing the recharge/discharge effect (Jin, 1997) due to the changes in wind stress curl instead of the Bjerknes feedback. We want to clarify that the effect of wind stress curl change actually acts to recharge the ocean heat towards the equatorial Pacific, which makes the local thermocline deeper. A more detailed discussion can be found in our response to the comment above.

Discussion: What is the impact of systematic model errors in the mechanism presented in this study? I think the authors need to discuss (and analyze) the seasonality of the response to determine whether it arises from the double ITCZ error or not.

Thanks for raising this important question. Model biases in the tropical Pacific mean state can potentially affect the model performance in the warming pattern. Following the reviewer’s suggestion, we analyzed the seasonality of the warming pattern and found that the eastern Pacific warming induced by wind-driven ocean circulation changes (i.e., FC-MD) is a robust feature found in all four seasons (Fig. R4). We also analyzed the model performance in mean-state SST and precipitation (Fig. R5). CESM2 has a reasonably good representation of the

single eastern Pacific ITCZ. In fact, the reduction of double ITCZ is one of the main improvements from CESM1 to CESM2, as highlighted in Danabasoglu et al. (2020). However, we agree with the reviewer that model mean state bias is a critical issue. For example, CESM2 has a generally too warm and thus too wet tropical Pacific. Other climate models may still suffer from the double ITCZ bias, among many other kinds of biases (e.g., cold tongue bias). These model mean-state biases can potentially downgrade the model performance in the warming pattern and need to be carefully investigated in future studies. We have now discussed the role of model bias in the last paragraph of the main text (Lines 217-229).

Fig. R4 | Seasonal variations of the tropical Pacific SST trend in FC-MD for boreal spring (February-April), summer (May-July), fall (August-October), and winter (November-January), respectively. Stippling in the panel indicates significance at the 95% confidence level.

Fig. R5 | The 1979–2014 mean-state SST (K) and mean-state precipitation (mm/day). The left panels show the observations from the ERSSTv5 SST dataset, FC SST, and the FC minus observations difference distributions, respectively. The right panels show the observations from the Global Precipitation Climatology Project (GPCP), FC precipitation, and the FC minus observations difference distributions, respectively.

Fig. 3. Something is odd about some labels in Fig. 3. I don't understand why 4 panels are shown for each advective term. I can imagine showing 3 panels, what is the 4th one? Also, the labels do not seem straightforward to understand. Please revise so it is easier to digest what each map is showing.

The labels in our previous version of Fig. 3 are indeed confusing. In Fig. 3, the left column is for three dominant advection terms, and the right column is for their own associated fields in ocean current and temperature. In other words, the 4 panels from b to e represent two advective terms in the meridional direction and their associated ocean current and temperature fields. To avoid the confusion, we have revised and renamed the labels shown in Fig. 3c, e, g, changing “ $-v^t \frac{\partial \bar{T}}{\partial y}$,”

to “ v^t (colors) and \bar{T} (contours)”, “ $-\bar{u} \frac{\partial T^t}{\partial y}$ ” to “ \bar{u} (colors) and T^t (contours)”, and “ $-\bar{v} \frac{\partial T^t}{\partial x}$ ” to “ \bar{v} (colors) and T^t (contours)” for clarity. We have also revised the figure caption to clarify (Lines 628-644).

References

1. Seager, R. et al. Strengthening tropical Pacific zonal sea surface temperature gradient consistent with rising greenhouse gases. *Nat. Clim. Change* **9**, 517–522 (2019).
2. Seager, R., Henderson, N., & Cane, M. Persistent discrepancies between observed and modeled trends in the tropical Pacific Ocean. *Journal of Climate*, **35**, 4571-4584 (2022).
3. Jin, F. F. An equatorial ocean recharge paradigm for ENSO. Part I: Conceptual model. *J. Atmos. Sci.* **54**, 811–829 (1997).
4. Danabasoglu, G. et al. The Community Earth System Model version 2 (CESM2). *J. Adv. Model. Earth Syst.* **12**, e2019MS001916 (2020).

Reviewer #2 (Remarks to the Author):

This study investigates the historical tropical Pacific warming pattern since 1958, characterized by warming in the eastern and western Pacific and cooling in the central tropical Pacific, as well as a hemispheric asymmetry in the subtropical eastern Pacific. Authors use the mechanical decoupled simulations to isolate the impact of wind-driven ocean circulations, which allow them to investigate the impact of dynamic and thermodynamic factors on the historical tropical warming pattern. They find that eastern tropical Pacific warming is related to southward Ekman transport induced by anomalous off-equatorial westerly winds. While the warming in the western tropical Pacific is attributed to heat transported by asymmetric equatorial surface winds from the northeastern Pacific, induced by historical external forcing. This paper is well written and provides important implications for the mechanisms of the observed tropical Pacific warming, particularly for the eastern tropical Pacific warming. Their proposed mechanism differs from previous studies, which suggested that the warming in the eastern tropical Pacific is caused by the deepening of the thermocline as a result of Bjerknes feedback. Overall, I think this paper is suitable for the scope of Nature Communications and I would recommend the acceptance of this manuscript once the suggested comments have been adequately addressed.

I have provided detailed comments below in my major and specific comments. My main concern is that authors highlight the observed tripolar Pacific warming pattern, including the weak cooling in the central tropical Pacific, however, authors don't give sufficient elaboration on the results about central Pacific in observations and models. Main text focus on western and eastern tropical Pacific warming. All bar charts include data for the central tropical Pacific, but the main text does not provide a comprehensive description or discussion of these results. Also, I would suggest authors to provide more information on the

robustness of SST trends across observation datasets, especially SST trends over central tropical pacific.

Thanks for the encouraging comments and constructive suggestions. We have now attempted to address this major concern and other specific comments. Please find our detailed reply below.

Major Comments:

1. The manuscript prominently highlights the observed tripolar warming pattern. However, models fail to reproduce the observed weak cooling in the central tropical Pacific, and authors do not provide a comprehensive explanation for results of central tropical Pacific shown in the figures. Main text focuses on explaining the causes of symmetric warming in the western tropical Pacific and asymmetric warming in the eastern tropical Pacific. There hasn't been sufficient explanation regarding the possible cause of the observed central tropical Pacific cooling (the tripolar warming pattern) and the discrepancy between models and observations. If the main focus is on the western and eastern tropical Pacific warming pattern, authors might consider adjusting the related context. Another point is that the time span of model simulations and observations are not the same. It is better to be consistent with the study period.

We agree with the reviewer that the model-observation mismatch in equatorial CP cooling is a critical issue that needs further discussion and investigations. We have now elaborated our discussion on this mismatch for CESM2 and most CMIP6 models and the potential connection to model mean-state biases (Lines 108-114, Lines 217-229). We have decided to leave it as an open question and clarified that the main focus of this paper is to be WP and EP warming (Lines 114-116). A few more discussions on the CP SST changes are added (Lines 217-227).

We compared the observed SST trends in 1958-2014 and 1958-2022, and we found consistent results (Fig. R5). We decided to use the period of 1958-2014 consistently for all our observational and modeling analysis but didn't update that information in the main text. We apologize for the confusion due to the typo and have now corrected that in the revised manuscript. We chose 1958 as the starting year by following Seager et al. (2019, 2022), which was determined according to the availability of ECMWF/ORAS4 ocean reanalysis data. We chose 2014 as the ending year because our CESM2 historical fully coupled and mechanical decoupled experiments both followed the standard of the CMIP6 historical simulations that ended in the year 2014 (Lines 239-244).

Fig. R5 | a-b The mean observed SST trend from five observations-based products over 1958–2014 and 1958–2022, respectively. Stippling indicates significance at the 95% confidence level.

Specific Comments:

1. In Fig.1a, the cooling over the central tropical Pacific and southeastern Pacific appears weak and statistically insignificant. Do all observation datasets show this cooling? Also, considering that the spatial resolution of observation datasets varies, it would be helpful to clarify whether all datasets are interpolated to the same resolution. Because interpolation could impact the estimate of observed SST trends and, therefore, their average, especially when dealing with datasets like Kaplan Extended SST v2, which has a coarser grid compared to others. It would be helpful to show SST trend maps for each observation dataset. An alternative way is to show the agreement in the sign of SST trends among observation datasets. Maybe include more observation datasets can increase confidence on the sign of SST change over central tropical Pacific.

Thanks for your suggestion. We plotted the trends in tropical Pacific SST from 1958 to 2014 using five different observational datasets and their average (Fig. R6). We found that all the observational datasets showed a cooling signal in the central Pacific, but they differ in the exact magnitude and structure. This cooling feature is consistent across the datasets investigated here but is not statistically significant due to the weak signal. Concerning the resolution issue, yes, we interpolated all observed data sets to the same grid to calculate the observed mean over multiple sets of observations. Given the broad agreement across datasets on the general structure of SST trends, we think the warm-cold-warm tripolar structure is a robust feature not influenced by the regridding process. This new figure is now included in the Supplementary Information as Supplementary Fig. 1.

Fig. R6 | The SST trend from five observations-based products over 1958–2014. Shown are those from (top left) HadISST, (top right) COBE, (middle left) COBE2, (middle right) ERSSTv5, (bottom left) Kaplan, and (bottom right) the mean of these five. Stippling indicates significance at the 95% confidence level.

2. Is the result sensitive to how the eastern and western tropical Pacific regions is defined? In Fig. 1c, the weaker warming over eastern tropical Pacific extends to central tropical Pacific in CEMS2 LENS. Also, the warming over eastern tropical Pacific extends to central tropical Pacific driven by externally forced winds in Fig. 1e.

It is a great question. To assess the sensitivity of our results to region selection, we tried different ways to define the range of boxes and found that the main conclusions remained the same. Below we show one example. We tried using narrower boxes (Fig. R7), that is, the Western Pacific (WP; 5°S-5°N, 120°E-160°, blue box), the Central Pacific (CP; 5°S-5°N, 180°-130°W, green box) and Eastern Pacific (EP; 5°S-5°N, 110°W-80°W). We found that the heat budget results were broadly consistent with those presented in the main text, although the exact numbers might be different as expected. This suggests our main conclusion is not sensitive to the definition of boxes. To show the full picture on

which our conclusions are based, we also showed the associated spatial structure of each budget term in Supplementary Fig. 2, Supplementary Fig. 3, and Supplementary Fig. 5.

Fig. R7 | **a**, Schematic of the chosen region, the Western Pacific (WP; 5°S-5°N, 120°E-160°, blue box), the Central Pacific (CP; 5°S-5°N, 180°-130°W, green box) and Eastern Pacific (EP; 5°S-5°N, 110°W-80°W). **b**, Bar chart showing the contribution of FC-MD SST trend from each component in the ocean mixed layer heat budget. **c**, Bar chart showing the diagnosed results of FC-MD advective terms in the oceanic mixed-layer heat budget. **d**, Same bar chart as **b**, but for MD.

3. Line 68 has a superscript stop.

Revised (Line 68).

REVIEWERS' COMMENTS

Reviewer #1 (Remarks to the Author):

Thanks for considering my suggestions in the review. The manuscript has improved in the process and it is ready for publication now.

Reviewer #2 (Remarks to the Author):

The authors have addressed my comments and discussed the model-observation discrepancy in the central tropical Pacific. According to their findings, CESM2 simulates a weaker climatological upwelling in the tropical Pacific, leading to a warmer central Pacific compared to observations. CMIP6 models also fail to capture the observed warm-cool-warm pattern in this region. CMIP6 models might potentially have large variations in their performance over the tropical Pacific. It would be interesting to investigate whether CMIP6 models that simulate an SST pattern closely resembling the observed pattern (with a relatively high spatial correlation with observed pattern), would have a stronger upwelling in the tropical Pacific than models with lower spatial correlation. This analysis can be time-consuming and can be potentially developed as a separate question on model biases in the central tropical Pacific.

Overall, I would recommend accepting this manuscript.

Best,

A reviewer for Nature Communication